# Analyzing the innovation ability of listed companies in the core area of the Huaihai economic zone

Qin-Xia Liu [iD] *

School of Economics and Management, Jiangsu Vocational Institute of Architectural Technology, Xuzhou, Jiangsu, China

* lqxia2004@163.com, 872969467@qq.com

## Abstract

The strategic position of the core area of the Huaihai Economic Zone is very important. The evaluation and analysis of the listed companies' innovation ability in this core area effectively reflect the level of innovation ability of regional enterprises and uncover the differences and influencing factors of the enterprise innovation ability level across different cities and industries; this would provide a reference for further improving the enterprise innovation ability level in the Huaihai Economic Zone. Given this context, data are collected from the CSMAR database on 37 listed companies in eight cities in the Huaihai Economic Zone core area from 2017 to 2021, and an innovation ability evaluation index is constructed from the innovation input and innovation output dimensions of listed companies. The results show that the innovation ability of listed companies in the region is weak; the main reason for the lack of innovation ability of listed companies is the lack of capital investment and talent investment; the innovation primacy of Xuzhou listed enterprises is not high. Finally, in view of the improvement of the innovation ability of listed enterprises in the core field, corresponding suggestions are put forward from the aspects of increasing innovation investment, optimizing the innovation environment and improving the innovation leading force in Xuzhou.

## Introduction

Innovation ability represents the core competitiveness of enterprise development and is the foundation of enterprise sustainable development. Through the comprehensive evaluation of enterprise innovation ability, we can measure the quantity and quality of enterprise innovation R&D investment and innovation output, further clarify the elements of enterprise innovation, and provide objective and comprehensive data support for the research and implementation of innovation policy [1–3]. As a group of Chinese enterprises with relatively sound mechanisms and systems, reasonable asset structures, and relatively transparent and open financial systems, listed companies are the leaders and navigators of the industry. They can promote the adjustment of industrial structure and the overall competitiveness of domestic enterprises; furthermore, they are the backbone of national economic growth [4]. For a region, the quantity

**Funding:** This work was supported by the Xuzhou Social Science Research Project, No. 22XSM-167,

Study on benchmarking gap and promotion path between Xuzhou and national central cities; Project of Jiangsu Vocational Institute of Architectural Technology, No. JYA320-24, Evaluation and comparative study on the primacy of scientific and technological innovation in Xuzhou from the perspective of value chain. The funders had no role in study design, data collection and analysis, decision to publish, or preparation of the manuscript.

**Competing interests:** The authors have declared that no competing interests exist.

and quality of listed companies can reflect the scale and strength of a regional economy. The innovation ability development level of listed companies also determines the innovation ability of the region to a certain extent.

The Huaihai Economic Zone is an economic cooperation zone at the junction of Jiangsu, Shandong, Henan and Anhui provinces, with Xuzhou as the center. It is also an economic growth zone between the Yangtze River Delta and Beijing-Tianjin-Hebei. In this region, the strategic position of the core area of the Huaihai Economic Zone is very important, as it plays an important role in connecting the East and the West and echoes the North and the South, including Xuzhou, Jining, Zaozhuang, Lianyungang, Suqian, Shangqiu, Suzhou and eight cities in Huaibei. Promoting the innovation ability of enterprises in the Huaihai region, promoting the transformation of innovation ability into development strength, and continuously providing power and innovation sources for the development of the Huaihai Economic Zone are very important research topics.

Based on data from 37 listed companies in eight cities in the Huaihai Economic Zone core area from 2017 to 2021, this paper uses the entropy method to construct the innovation ability evaluation system of these companies, determines the corresponding indicators and weights, calculates and evaluates the innovation ability across these companies, and compares and analyzes the differences and influencing factors in terms of their innovation ability level, thus providing a reference for further improving the innovation ability of enterprises in the Huaihai Economic Zone.

## Literature review

The research of foreign scholars on enterprise innovation ability started relatively early, and its results are very rich. There is much research on the connotation of enterprise innovation ability, the evaluation of enterprise innovation ability and the influencing factors of innovation ability, mainly including Schumpeter [5], who believes that enterprise innovation is determined by enterprise scale and market power. The innovation capability of a country is closely related to that of an enterprise. Leonard-Barton [6] believes that core enterprise technological innovation includes three aspects: enterprise values, employees with professional knowledge, and capabilities related to technology systems and management systems. Lesáková [7] evaluated the innovation capacity of SMEs in Slovakia from the aspects of economic growth and development and competitiveness and policies supporting innovation. Forsman [8] found that innovation models in small manufacturing and service industries are diversified and that there should be direct policies for supporting the innovative development of small enterprises. Crepesca [9] systematically studied the evaluation of enterprise innovation ability and its influencing factors, and Dodgson et al. [10] systematically discussed the evaluation of enterprise innovation ability and performance. Feniser et al. [11] used TRIZ theory to analyze the sustainable development of small and medium-sized enterprises and formed an evaluation flow chart of small and medium-sized enterprise innovation. Lee et al. [12] analyzed the technological innovation ability of SMEs and the influencing factors of the technological innovation activities of SMEs. Greco et al. [13] found that there is a strong correlation between technological innovation efficiency and government subsidies by studying the technological innovation efficiency of enterprises.

In recent years, Chinese scholars have conducted abundant research on the innovation ability of listed enterprises. Among them, many scholars have comprehensively evaluated the innovation ability of enterprises. For example, Chen and Xu [14] compared and evaluated the innovation ability of 65 listed companies in Jiangsu Province using the analytic hierarchy process (AHP). Sun [15] used a factor analysis method to evaluate and study the innovation

capability of 36 listed companies in Nanjing and found that there were differences in the allocation of innovation R&D investment resources of these companies across different industries and that the size of enterprises affected the levels of enterprise innovation capability. Many scholars have also conducted in-depth research on the influencing factors of listed enterprise innovation capability. For example, Zhang et al. [4] studied the innovation capability of 41 listed companies in Shaanxi Province by using a structural equation model and principal component analysis. It was found that the industry nature, innovation input and innovation driving factors of all listed companies have an important influence on innovation ability. Xiao and Liu [16] evaluated and analyzed the innovation ability of 56 listed agricultural enterprises in China and found that the main factors affecting the innovation ability of these enterprises include enterprise scale and finance, R&D output, regional innovation ability, R&D input, personnel quality and enterprise growth. At the same time, the innovation ability of listed agricultural enterprises across different provinces is obviously different. Many other scholars have analyzed and studied the correlation between innovation capability and other factors. For example, by constructing an innovation capability model, Xu et al. [17] conducted an empirical study on 82 listed companies in the science and technology service industry and analyzed the impact of foreign direct investment on the innovation capability of these companies. It is found that the scale and efficiency of listed companies are positively correlated with the innovation ability of enterprises. Feng et al. [18] used the entropy weight-TOPSIS model to evaluate the innovation ability of China's listed coal enterprises and found that investment in technological innovation has the greatest impact on the development of the technological innovation ability of listed coal enterprises.

Generally, Chinese scholars' research on the innovation ability of listed companies is relatively comprehensive, and it includes analysis from the national, provincial, regional, and industry perspectives. The analysis methods used in studies are very rich, including factor analysis, cluster analysis, analytic hierarchy process, and the entropy method. However, there is no relevant research on the evaluation and analysis of the innovation ability of listed companies in the Huaihai Economic Zone. Therefore, this paper takes the listed companies in the core area of the Huaihai Economic Zone as the research object and uses the entropy method for evaluation. Using the Boston matrix analysis method, the innovation input, innovation output and input–output ratio of all listed companies are compared to explore the advantages, gaps and influencing factors of the innovation ability of such companies. To analyze and evaluate the innovation ability of listed companies in the region more comprehensively, the entropy-TOPSIS model is also used to evaluate the innovation ability of listed companies in the region. The paper complements and enriches the research with respect to the innovation ability of listed enterprises in the Huaihai Economic Zone and proposes corresponding countermeasures and suggestions to provide a reference for improving the innovation ability of the region. The research framework of this paper is shown in Fig 1.

## Methodology

### Study design

**Sample selection.** The research data in this paper are derived from the CSMAR database. Taking the listed companies in the core area of the Shanghai and Shenzhen A-share Huaihai Economic Zone as the research sample, excluding the ST shares and the enterprises with missing relevant index data, a total of 37 companies were selected, and the 2017–2021 enterprise data were selected as the research object.

**Construction of the innovation capability evaluation index system.** Considering that innovation input and output are the preconditions for objectively and scientifically measuring

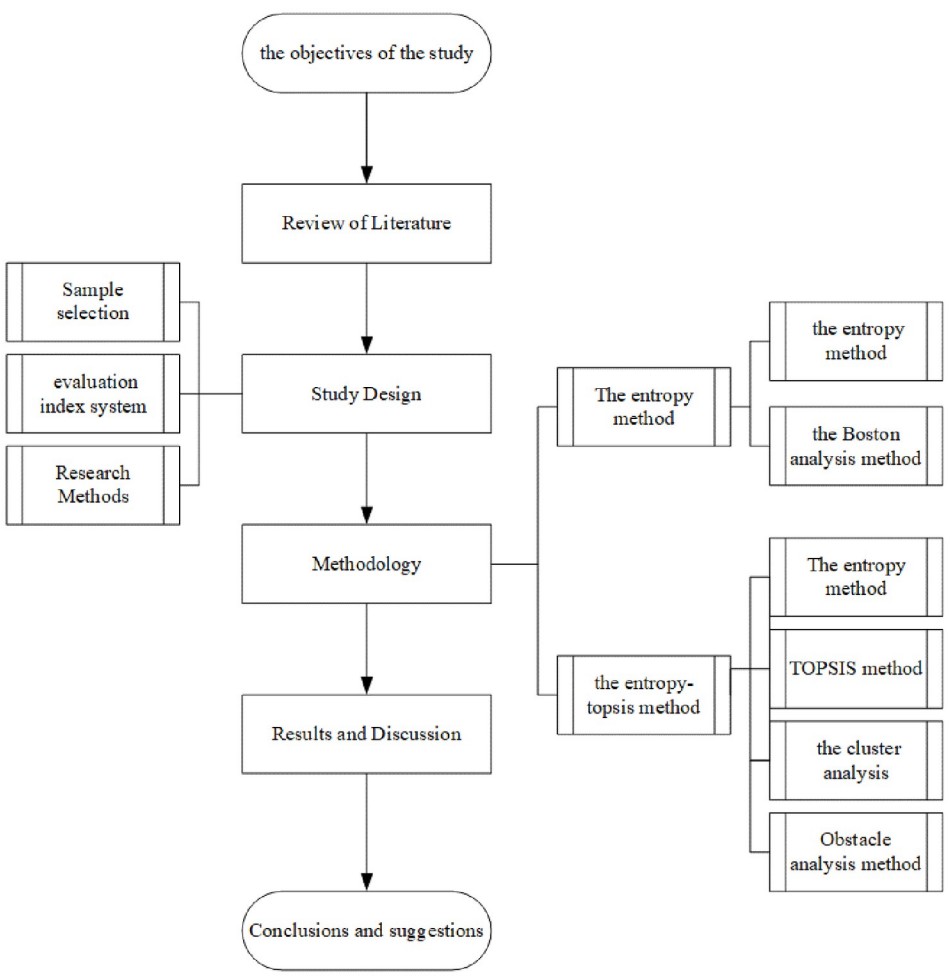

**Fig 1. The research technology roadmap.**

regional innovation capability, this paper constructs an innovation capability evaluation index system from the two dimensions of "innovation input" and "innovation output" of listed companies [19]. Specifically, 12 indicators are selected to measure the innovation ability of enterprises, and these indicators cover the process of innovation input and the outcome of innovation output. From the perspective of the innovation input process, six indicators are selected: total assets, R&D investment intensity, R&D investment amount, R&D personnel proportion, R&D personnel number, and the proportion of employees with a bachelor's degree or above. From the perspective of innovation output results, six indicators [19, 20] are selected, including intangible assets, growth rate of intangible assets, growth rate of revenue, growth rate of net profit, return on equity and gross operating margin. The results are shown in Table 1.

## Research methods

**Overview of the method.** This paper uses a variety of methods to comprehensively evaluate and analyze the innovation ability of listed companies in the core area of the Huaihai Economic Zone. First, the entropy method is used to evaluate the innovation ability of listed companies in the region, and the Boston analysis method is used to analyze the innovation

**Table 1. The index screening results.**

| First grade index | Second grade index | Third grade index | Computing formula | Index types |
|---|---|---|---|---|
| Research on the innovation ability of different listed companies in core area of Huaihai economic zone | Innovation input | Total assets | Scale of total assets of enterprises on balance sheet date | Positive index |
| | | R&D investment intensity | R&D investment amount/operating income | Positive index |
| | | R&D investment | Amount of R&D investment disclosed in financial statements | Positive index |
| | | Proportion of R&D personnel | Number of R&D staff/total staff | Positive index |
| | | Quantity of R&D personnel | Number of R&D personnel disclosed in financial reports | Positive index |
| | | Proportion of employees with Bachelor's degree or above | Number of employees with bachelor's degree or above/total number of employees | Positive index |
| | Innovation output | Intangible assets | Amount of intangible assets on balance sheet date | Positive index |
| | | Growth rate of intangible assets | Increase in intangible assets/amount of intangible assets in the same period last year | Positive index |
| | | Income growth rate | Income increase/income of the same period last year | Positive index |
| | | Net profit growth rate | Net profit increase/net profit in the same period last year | Positive index |
| | | ROE | Net profit/average net assets | Positive index |
| | | Gross operating margin | (Income minus cost)/income | Positive index |

input, innovation output, innovation input–output ratio score and influencing factors of listed companies in different industries and cities. To further verify the conclusion of the entropy method and to more comprehensively analyze and evaluate the innovation ability of listed companies in the region, the entropy-topsis analysis method is used to evaluate the innovation ability of listed companies in the region, and cluster analysis is carried out. At the same time, the factors affecting the improvement of the innovation ability of listed companies are analyzed by the obstacle degree analysis method.

**The entropy method.** The entropy weight method is an objective weighting method. It is believed that the greater the degree of data change involved in a system, the greater the relative weight of the index. The weight coefficient of each index is calculated by information entropy [21–24]. The steps of the entropy weight method are as follows:

1. Index standardization

The positive indicators are standardized by the following formula.

$$V'_{ij} = {}^{V_{ij}-MIN(V_{ij})}\big/_{MAX(V_{ij})-MIN(V_{ij})} \tag{1}$$

and the negative indicators are standardized by formula 2

$$V'_{ij} = {}^{MXA(V_{ij})-V_{ij}}\big/_{MAX(V_{ij})-MIN(V_{ij})} \tag{2}$$

Among them, $1 \leq i \leq n$, $1 \leq j \leq m$, n is the number of samples of listed companies, and m is the number of evaluation indexes.

2. Calculate the information entropy of the evaluation index

$$E_j = -k \sum_{i=1}^{n} f_{ij} \times ln f_{ij}, i = 1, 2, \ldots, n \qquad (3)$$

where $k = \frac{1}{ln\,n}$; $f_{ij} = \frac{V'_{ij}}{\sum_{i=1}^{n} V'_{ij}}$; and $0 \leq E_j \leq 1$. When $f_{ij} = 0$, it is assumed that $lim\, f_{ij}\, ln\, f_{ij} = 0$.

3. The weight of each index is calculated, as shown in the formula.

$$W_j = \frac{G_j}{\sum G_j}, j = 1, 2, 3 \ldots, m \qquad (4)$$

where m is the number of evaluation indexes and $G_j = 1 - E_j$.

**Topsis method.** The basic principle of the TOPSIS method is to construct the positive ideal solution and negative ideal solution of multiobjective decision-making and evaluate the advantages and disadvantages of multiobjective decision-making according to the Minkowski distance from the evaluation object to the positive and negative ideal solutions. The Euclidean geometric distance is a special case of the Minkowski distance [21–24]. The application process of the TOPSIS method is as follows:

1. The evaluation object weighted evaluation matrix

$$r_{ij} = (W_j * V'_{ij})_{(n*m)} \qquad (5)$$

2. By calculating the positive and negative ideal solutions.

the Euclidean distance from different decision schemes to $R^+$ is denoted by $s^+$, and the Euclidean distance from $R^-$ is denoted by $s^-$, then:

$$s^+ = \sqrt{\sum_{j=1}^{m} (r_{ij} - r_j^+)^2} \qquad (6)$$

$$s^- = \sqrt{\sum_{j=1}^{m} (r_{ij} - r_j^-)^2} \qquad (7)$$

3. The comprehensive score of the fitting degree is calculated.
   The closeness degree Ci + (i = 1, 2... n) to the positive ideal solution is calculated. A larger Ci + indicates that the decision is closer to the positive ideal solution. A smaller Ci + indicates that the decision is closer to the negative ideal solution.

$$C_i = \frac{s_i^-}{s_i^- + s_i^+} \qquad (8)$$

**Obstacle model.** Using the obstacle degree model, the three indicators of factor contribution, index deviation and obstacle degree are used to analyze the obstacle factors that hinder the development of the innovation ability of listed companies in the core area of the Huaihai Economic Zone [21–24].

1. Factor contribution.
   Factor contribution ($W_j$) is the contribution of a single factor to the overall goal, that is, the weight of a single factor.

2. Index deviation.
   The index deviation ($D_{ij}$) is the gap between the single factor index and the system development goal, that is, the difference between the standardized value of the single index and the 100% target:

$$D_{ij} = 1 - Y_{ij} \qquad (9)$$

3. Obstacle degree.
   The obstacle degree ($O_i$) is the influence degree of a single index on the innovation ability of listed companies in the core area of the Huaihai Economic Zone. The greater the obstacle degree, the higher the degree of restricting the further development of the innovation ability of listed companies.

$$O_j = \frac{D_{ij} * W_j}{\sum_{j=1}^{m} (D_{ij} * W_j)} \qquad (10)$$

## Results and discussion

### Evaluation results and discussion of the entropy method

**The evaluation index weight of the entropy method.** First, this paper evaluates the innovation ability of listed companies in the core area of the Huaihai Economic Zone by the entropy method. According to the entropy method, the index weight of the innovation ability index evaluation system constructed in this paper can be determined, as shown in Table 2.

**Innovation capability evaluation of listed companies.** According to the index weights obtained above, the strength of the innovation ability of listed companies in the core area of the Huaihai economic zone can be calculated. The specific process is as follows. According to the calculated weight of each index, the comprehensive score of the company's innovation ability is calculated. The score is the evaluation result of the company's innovation ability. Finally, the enterprise innovation ability level is divided according to the score.

According to the calculation results, companies raised their overall innovation input, and the innovation output results also improved, although the innovation input and innovation output of most companies fluctuated in the period of 2017 to 2021. However, from the perspective of the input–output ratio, the improvement in innovation output was too small, and the input–output ratio of the vast majority of enterprises declined to varying degrees due to the impact of the epidemic after 2019.

Based on the average calculation of the innovation input, innovation output and input–output ratio of each company from 2017 to 2021, it is found that the top 10 innovation inputs are Yunyi, Hengrui, Huaibei Mining, Yankuang, Lianyungang, XCMG, Grand Holding, Hengyuan, Sun Paper and Yabo; the top 10 innovation outputs are Hengrui, Huaibei Mining, KOUZI, Yanghe, Kanion, Nhwa, Cisen, Hengyuan, Novoray and Yankuang; and the input–output ratios of the top 10 are KOUZI, VVFB, Liancheng, LFBC, Yanghe, Nhwa, Solareast, Huaxin, Novoray, and Ruyi. The detailed results are shown in Table 3.

**Table 2. The evaluation index weight.**

| First grade index | Second grade index | Weight of second grade index | Third grade index | Weight of third grade index |
|---|---|---|---|---|
| Research on the innovation ability of different listed companies in core area of Huaihai economic zone | Innovation input | 44.21% | Total assets | 6.66% |
| | | | R&D investment intensity | 9.63% |
| | | | R&D investment | 3.07% |
| | | | Proportion of R&D personnel | 9.89% |
| | | | R&D personnel | 7.96% |
| | | | Proportion of employees with Bachelor's degree or above | 7.00% |
| | Innovation output | 55.79% | Intangible assets | 2.76% |
| | | | Growth rate of intangible assets | 10.50% |
| | | | Income growth rate | 8.45% |
| | | | Net profit growth rate | 11.40% |
| | | | Rate of return on net assets | 11.43% |
| | | | Gross operating margin | 11.24% |

To further compare the differences in the innovation capabilities of listed companies in the Huaihai economic core area, a Boston matrix was drawn for the average innovation input and innovation output of each company from 2017 to 2021 (Fig 2), where the abscissa represents the urban innovation output, and the ordinate represents the urban innovation input. With a mean innovation input value of 3.52 and a mean innovation output value of 23.81 as the boundary of the quadrant, the innovation input and output of 37 cities in the Huaihai economic core area can be divided into four categories.

First, the high input and high output quadrant includes Hengrui, Huaibei Mining, Hengyuan and Yankuang. The quadrant includes one pharmaceutical company, and the other three are coal enterprises. This shows the contribution of innovation input to the development of enterprises. The epidemic has brought a large number of business opportunities to the development of pharmaceutical enterprises, but enterprises that give importance to innovation input can better grasp the opportunity due to the accumulation of technology. Due to long-term innovation input, the three coal mining enterprises seized the golden opportunity for the development of coal mining technology due to the world energy shortage.

Hengrui Medicine is a typical representative of excellent enterprises with high input and high output. Hengrui Pharmaceutical is a leading enterprise in China's pharmaceutical industry. The development of Hengrui Pharmaceutical is mainly due to the innovative transformation of the early layout and the continuous high-intensity investment in research and development. R&D investment has increased rapidly from 400 million yuan in 2011 to 6.203 billion yuan in 2021. At the same time, the continuous output of the company's innovative products, the upgrading of product innovation and the expansion of diversified products led the company's R&D output to increase rapidly.

The second is the high input low-output quadrant, which includes companies Yunyi and Lianyungang, XCMG, Grand Holding, Sun Paper, Yabo, Saimo and Sidike. Among them, XCMG and Yunyi belong to the manufacturing industry, which has had a long input–output cycle. The conflict of the epidemic on related industries led to high input and low output.

**Table 3. The average innovation ability evaluation results of enterprises in the Huaihai economic core area from 2017 to 2021 (company English Name for Brevity).**

| Company | Innovation input | Innovation output | Innovation input–output ratio | City | Classification of occupations |
|---|---|---|---|---|---|
| KOUZI | 0.7169 | 27.8573 | 39.5372 | Huaibei | Intoxicating liquor |
| VVFB | 2.381 | 22.6269 | 15.763 | Xuzhou | Food and drink |
| Liancheng | 2.1733 | 22.7918 | 14.7796 | Jining | Auto parts |
| LFBC | 1.4304 | 18.5331 | 14.4228 | Xuzhou | Pesticide veterinary drugs |
| Yanghe | 2.0225 | 27.5431 | 14.1142 | Suqian | Intoxicating liquor |
| Nhwa | 2.276 | 26.6881 | 13.2916 | Xuzhou | Chemical pharmacy |
| Solareast | 1.7393 | 22.9153 | 13.2456 | Lianyungang | Household appliances |
| Huaxin | 1.7722 | 22.5119 | 12.7152 | Xuzhou | Plastics |
| Novoray | 2.0108 | 25.3151 | 12.6301 | Lianyungang | Nonmetallic material |
| Ruyi | 2.1044 | 22.9853 | 11.176 | Jining | Textile and costume |
| Xiuqiang | 2.8994 | 22.7849 | 10.6984 | Suqian | Household appliances |
| Quartz | 2.3646 | 24.5912 | 10.5323 | Lianyungang | Nonmetallic material |
| Shenhuo | 2.2793 | 22.4832 | 10.2886 | Shangqiu | Forel |
| Kanion | 2.895 | 27.323 | 9.7965 | Lianyungang | Traditional Chinese medicine |
| Huaibei Mining | 9.4326 | 28.0395 | 9.6223 | Huaibei | Coal |
| Donghong | 2.3703 | 22.7092 | 9.6124 | Jining | Decorative materials |
| Shuangxing | 2.5185 | 22.5601 | 9.5271 | Suqian | Plastics |
| lukang | 2.4845 | 23.013 | 9.5172 | Jining | Chemical pharmacy |
| Sacred Sun | 2.4042 | 21.6721 | 9.448 | Jining | Cell |
| Fengyuan | 2.4249 | 21.4934 | 9.1753 | Zaozhuang | Chemical materials |
| Wuyang | 2.6633 | 23.1621 | 8.8758 | Xuzhou | Flexible unit |
| Taihe | 3.2413 | 23.8006 | 8.2558 | Zaozhuang | Chemical |
| Huafu | 2.9271 | 20.9762 | 7.1693 | Huaibei | Textile and costume |
| Limin | 3.4274 | 23.4741 | 6.9406 | Xuzhou | Pesticide veterinary drugs |
| Cisen | 3.7915 | 25.5268 | 6.8629 | Jining | Chemical pharmacy |
| Sidike | 3.8574 | 23.3708 | 6.5955 | Suqian | Plastics |
| Shantui | 3.5276 | 21.7441 | 6.1984 | Jining | Construction machinery |
| Saimo | 3.8645 | 22.9354 | 5.9883 | Xuzhou | Electric meter and instrument. |
| Sun Paper | 4.9053 | 22.9147 | 5.6194 | Jining | Paper printing |
| Grand Holding | 5.4979 | 20.6036 | 5.5011 | Lianyungang | Trade business |
| Hengyuan | 5.4484 | 25.5104 | 5.1143 | Huaibei | Coal |
| Yabo | 3.8918 | 17.3925 | 4.6916 | Zaozhuang | Decorative materials |
| Lianyungang | 7.2691 | 22.5857 | 3.5036 | Lianyungang | Shipping ports |
| XCMG | 6.7927 | 22.4495 | 3.3591 | Xuzhou | Construction machinery |
| Hengrui | 9.4824 | 28.9296 | 3.2565 | Lianyungang | Chemical pharmacy |
| Yankuang | 9.0016 | 24.879 | 2.793 | Jining | Coal |
| Yunyi | 10.6084 | 23.5257 | 2.236 | Xuzhou | Auto parts |

Lianyungang and Grand Holding, which belong to the port and logistics trade industries, have been significantly affected by the epidemic.

The third is the low input high output quadrant. The companies in this quadrant are KOUZI, Yanghe, Kanion, Nhwa, Cisen, Novoray, Quartz and Taihe. This quadrant mainly includes liquor enterprises and pharmaceutical enterprises. For liquor enterprises, the product process is more mature, the target users are more stable, and the input and output are relatively large. Pharmaceutical companies have seized the development opportunities brought by epidemic prevention and control, and their overall output has been good.

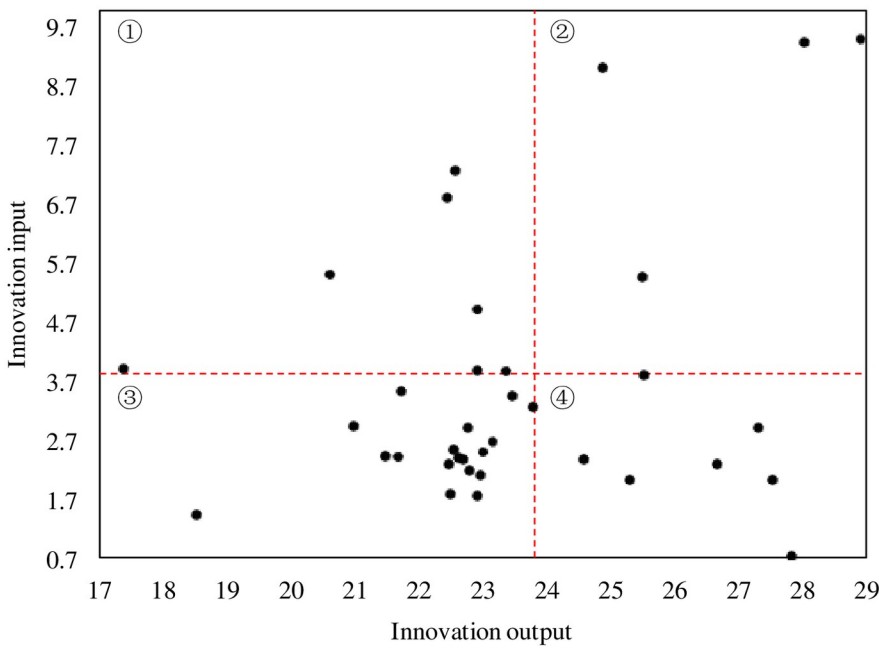

**Fig 2. The innovation input–output matrix diagram.** Note: ①-high input & low output; ②-high input & high output; ③-low input & low output; ④-low input & high output.

The fourth is the low-input, low-output quadrant. Companies in this quadrant include Limin, Wuyang, Lukang, Ruyi, Solareast, Liancheng, Xiuqiang, Donghong, VVFB, Shuangxing, Huaxin, Shenhuo, Shantui, Sacred Sun, Fengyuan, Huafu and LFBC; this quadrant mainly includes agricultural chemical products, chemical pharmaceuticals, equipment manufacturing and textile manufacturing, which are all traditional manufacturing enterprises. Such enterprises have limited demand for innovation input, and their development competitiveness is not outstanding, so it is difficult for them to become industry leaders with obvious advantages through innovation and development.

## Evaluation of urban innovation capability

The number of listed companies in the core of the Huaihai economic region is not balanced. Xuzhou and Jining had the largest number of listed companies, with 9 each. This is followed by Lianyungang with 8. Huaibei and Suqian each had 4. Then, in Zaozhuang city, there are 3. Finally, there is only one in Shangqiu city, and there is no listed enterprise in Suzhou city. See Table 4 for specific data.

**Table 4. List of the listed companies by city.**

| City | Different listed companies |
|---|---|
| Xuzhou | XCMG, Nhwa, LFBC, Limin, Yunyi, Wuyang, Saimo, Huaxin, VVFB |
| Jining | Shantui, Sun Paper, Ruyi, Sacred Sun, Liancheng, Yankuang, lukang, Cisen, Donghong |
| Lianyungang | Grand Holding, Hengrui, Kanion, Lianyungang, Solareast, Quartz, Novoray |
| Huaibei | Huafu, Hengyuan, Huaibei, KOUZI |
| Suqian | Yanghe, Shuangxing, Xiuqiang, Sidike |
| Zaozhuang | Yabo, Fengyuan, Taihe |
| Shangqiu | Shenhuo |

**Table 5. The score of the city innovation input.**

| Time | Huaibei | Jining | Lianyungang | Shangqiu | Suqian | Xuzhou | Zaozhuang |
|------|---------|--------|-------------|----------|--------|--------|-----------|
| 2017 | 1.871 | 3.0821 | 6.7265 | 1.9708 | 1.6081 | 3.7647 | 3.0568 |
| 2018 | 5.2224 | 3.3754 | 3.7958 | 1.9211 | 1.734 | 4.0557 | 3.6186 |
| 2019 | 5.2344 | 4.2234 | 3.8082 | 1.7846 | 3.0121 | 4.0428 | 3.7892 |
| 2020 | 5.2825 | 3.763 | 4.172 | 2.1458 | 3.0599 | 3.9186 | 2.7772 |
| 2021 | 5.546 | 3.7577 | 4.7539 | 3.574 | 3.615 | 3.7826 | 2.7526 |

The innovation ability of an enterprise is closely related to the innovation level of the economic ecosphere of the city where it is located, and the improvement in the innovation ability of an enterprise plays a leading role in the high-quality development of the city and can also measure the strength of the innovation ability of the region where it is located. The average innovation ability of listed companies in a city is taken as the index to measure the strength of the city's innovation ability; the final results are shown in Table 5 and Fig 3.

As shown in Table 5 and Fig 3, the innovation input of several cities in the core economic area of Huaihai showed an overall increasing trend from 2017 to 2021. However, some cities are in continuous decline, such as Zaozhuang city. With the highest score of innovation input in 2017, there was a significant decline in Lianyungang city from 2018 to 2021; the decline is manifested in 6.726 points in 2017 to 4.754 points in 2021. However, the overall score of its innovation input is still the highest. Lianyungang City has the highest average score of innovation input, mainly due to Hengrui. The largest increase in innovation input was in Huaibei city, whose score rose from 1.871 in 2017 to 5.546 in 2021, showing a significant increase. The overall score of Xuzhou innovation input is relatively stable, and the average score ranks third. As the Huaihai economic zone center city, Xuzhou has vigorously implemented a strategy of innovation in recent years, further promoting the reform of science and technology systems, fostering an innovative economy and technology innovation, and strengthening enterprise technology innovation ability as the first driving force leading development in Xuzhou.

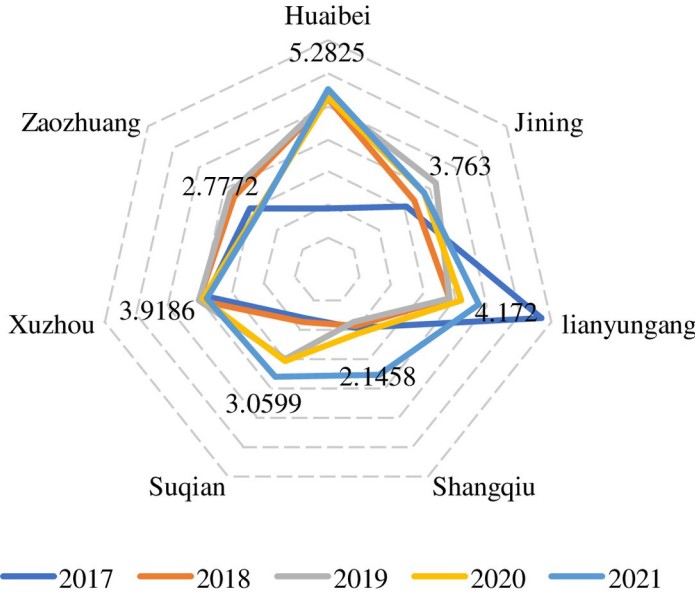

**Fig 3. The score chart of the city innovation input.**

**Table 6. Score of the city innovation output.**

| Time | Huaibei | Jining | Lianyungang | Shangqiu | Suqian | Xuzhou | Zaozhuang |
|---|---|---|---|---|---|---|---|
| 2017 | 25.7344 | 23.2079 | 24.4697 | 22.4599 | 23.8540 | 23.3721 | 23.7690 |
| 2018 | 30.4254 | 23.1169 | 24.0318 | 21.2658 | 24.0374 | 22.4262 | 17.7781 |
| 2019 | 24.1540 | 23.0614 | 24.7735 | 20.0465 | 23.6716 | 23.2160 | 20.1383 |
| 2020 | 23.4244 | 22.9640 | 24.4874 | 22.1972 | 24.5983 | 23.4351 | 20.2751 |
| 2021 | 24.2409 | 23.3365 | 24.9788 | 26.4467 | 24.4497 | 21.9433 | 20.4988 |

However, compared to a handful of other cities in the Huaihai economic zone, Xuzhou also needs to continue to increase investment in innovation.

It can be seen from Table 6 and Fig 4 that during the period of 2017 to 2021, for a few cities in the Huaihai economic zone, the trend of innovation output has also experienced a slight increase but has had different degrees of volatility. Xuzhou, Lianyungang, and Jining's overall innovation output increased slightly and was less volatile. Huaibei, Zaozhuang and Shangqiu were volatile. Among them, the output score of Huaibei was high in 2018, reaching 30.4254 points, then decreased significantly from 24.1540 points in 2019, and finally increased slowly to 24.24 points in 2021. However, the score of its overall innovation output was still relatively high. The innovation output score of Shangqiu city in 2017 was 22.4599 points, but it declined continuously from 2018 to 2019. However, it began to improve rapidly from 2019, and it increased significantly in 2021, reaching 26.4467 points in 2021.

From Table 7 and Fig 5, it can be seen that the ratio of innovation input–output of several cities in the core economic area of Huaihai showed a downward trend in most cities from 2017 to 2021, and the change fluctuated greatly. However, some cities showed a slow increase, such as Jining city and Zaozhuang city. With the largest decline in the innovation input–output ratio, Huaibei's score dropped from 13.7546 in 2017 to 4.3709 in 2021, which was a significant decline. The innovation input score of Suqian city in 2017 was the highest at

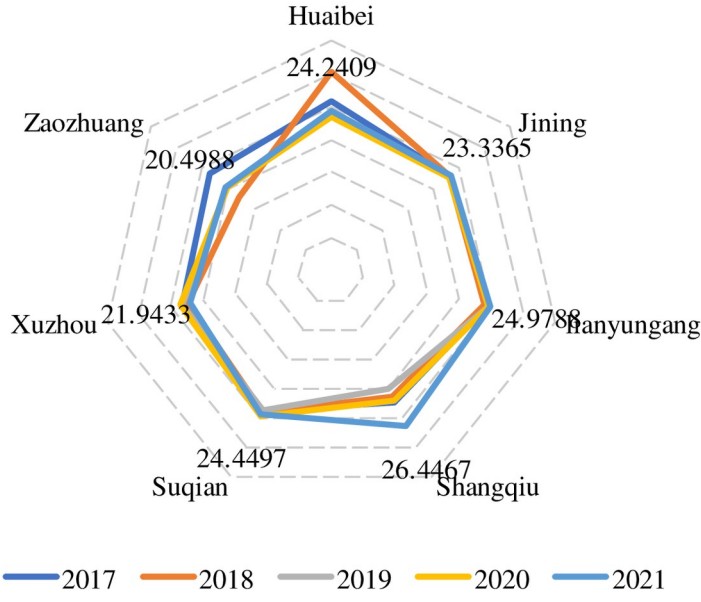

**Fig 4. Score chart of the city innovation output.**

**Table 7. The score of the city innovation input–output ratio.**

| Time | Huaibei | Jining | Lianyungang | Shangqiu | Suqian | Xuzhou | Zaozhuang |
|------|---------|--------|-------------|----------|--------|--------|-----------|
| 2017 | 13.7546 | 7.5299 | 3.6378 | 11.3961 | 14.8338 | 6.2082 | 7.7757 |
| 2018 | 5.8260 | 6.8487 | 6.3311 | 11.0694 | 13.8627 | 5.5295 | 4.9130 |
| 2019 | 4.6145 | 5.4604 | 6.5053 | 11.2331 | 7.8588 | 5.7426 | 5.3147 |
| 2020 | 4.4344 | 6.1025 | 5.8694 | 10.3444 | 8.0389 | 5.9804 | 7.3006 |
| 2021 | 4.3709 | 6.2104 | 5.2544 | 7.3998 | 6.7634 | 5.8011 | 7.4471 |

14.8338, and then it decreased all the way from 2017 to 2021 to 6.7634 in 2021, but overall, the innovation input–output ratio score was still relatively high from 2017 to 2021.

Based on the average calculation of the innovation input and output of each city from 2017 to 2021, the average value of innovation capability is used to comprehensively reflect the strength of the innovation capability of the city. As shown in Fig 6, there are obvious differences in the innovation capabilities of several cities in the core economic area of Huaihai. In terms of innovation input, Lianyungang and Huaibei ranked high with 4.65 and 4.63 points, respectively; however, Xuzhou and Jining were in the middle with 3.91 and 3.64 points, respectively, and Suqian and Shangqiu ranked low with 2.61 and 2.28 points, respectively. In terms of innovation output, Huaibei ranked first with 25.60 points, followed by Lianyungang and Suqian with 24.55 and 24.12 points, respectively, Jining and Xuzhou with 23.14 and 22.88 points, respectively, and Zaozhuang ranked last with 20.49 points. In terms of the innovation input–output ratio, Shangqiu and Suqian ranked first. Although the scores were 10.29 and 10.27, respectively, Shangqiu and Suqian did not have advantages because of the small number of listed enterprises. Huaibei and Zaozhuang were next with 6.60 and 6.55 points, respectively, followed by Lianyungang and Suqian, then Jining and Xuzhou, with Lianyungang ranking last with 5.52 points. It can be seen that the innovation ability of listed companies in each city of the Huaihai economic region is obviously different, the good and bad are uneven, and the number of listed companies with strong innovation ability is lower. At the same time, Xuzhou,

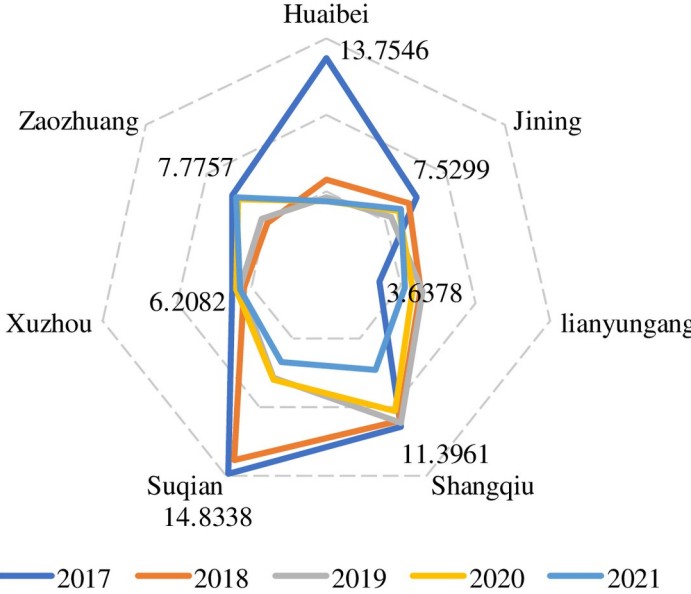

**Fig 5. The score chart of the city innovation input–output ratio.**

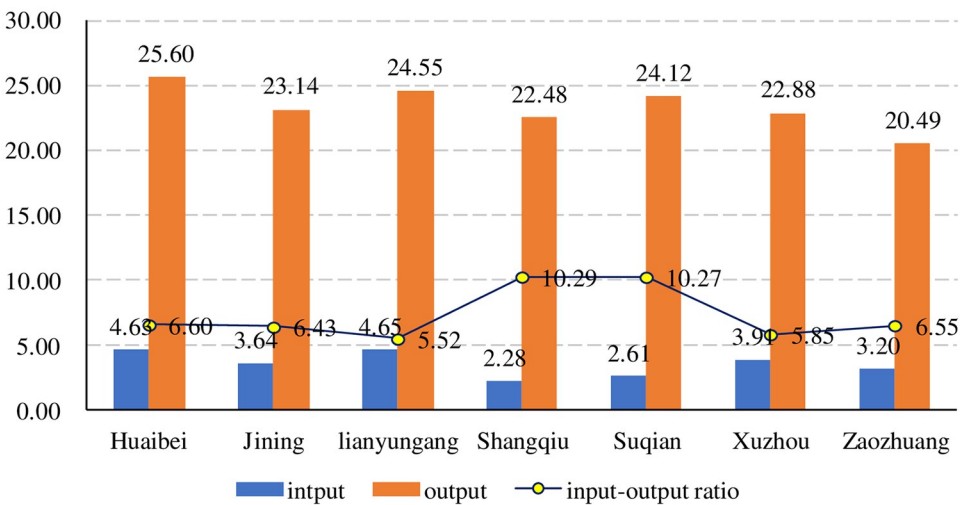

**Fig 6. The city average innovation input–output score.**

as the central city of the Huaihai region, has no obvious advantage in terms of innovation primacy.

Based on the previous analysis, compared with other cities, Huaibei, Lianyungang and Xuzhou are in the forefront of innovation ability, mainly because they have Huaibei Mining, Hengrui and XCMG as leading companies, respectively. The characteristics of these enterprises focus on scientific research and innovation, continuing to strengthen R&D investment, and maintaining strong R&D strength. They are typical representatives of high-quality innovation for promoting high-quality development. At the same time, the development of these enterprises is also related to the policy support provided by the local government and the creation of a good innovation environment. Taking Huaibei Mining as an example, Huaibei Mining is an excellent enterprise with high input, high output and a high input–output ratio. In recent years, Huaibei Mining has transformed from a traditional coal enterprise into an energy and chemical group with coal power, chemical industry and modern service industry as a leading industry by comprehensively promoting automation, intelligence, informatization and greening and comprehensively improving innovation ability. As the central city of the Huaihai Economic Zone, Xuzhou's listed companies' innovation ability is not high. The main reason is that its listed companies are mainly involved in traditional manufacturing industries. Therefore, it is necessary to further cultivate high-tech industries and develop emerging technologies so that Xuzhou's economic volume and industrial structure can play a central role in the Huaihai Economic Zone.

## Evaluation results and discussion of the entropy-topsis model

**The evaluation index weight of the entropy-topsis model.** The innovation ability of 37 listed companies in the core area of the Huaihai Economic Zone is dynamically changing. Therefore, to better show the dynamic evolution process of enterprise innovation ability, this paper divides the data of 37 listed companies in the core area of the Huaihai Economic Zone from 2017 to 2021 into years and uses the entropy weight method to obtain the weight of each year. Finally, the arithmetic average of the five-year weight value is used as the comprehensive weight of the innovation ability evaluation index of the listed companies in the core area of the Huaihai Economic Zone. The results are shown in Table 8.

**Table 8. Evaluation index weight of the innovation ability of listed companies.**

| index | 2017 | 2018 | 2019 | 2020 | 2021 | comprehensive weight |
|---|---|---|---|---|---|---|
| Total assets | 12.45% | 12.58% | 13.45% | 16.85% | 17.31% | 14.53% |
| R&D investment intensity | 9.43% | 3.75% | 3.34% | 3.97% | 4.58% | 5.01% |
| R&D investment | 30.51% | 14.66% | 15.22% | 18.31% | 18.76% | 19.49% |
| Proportion of R&D personnel | 5.28% | 5.02% | 4.25% | 5.21% | 5.06% | 4.96% |
| R&D personnel | 7.00% | 9.92% | 10.23% | 12.43% | 12.69% | 10.45% |
| Proportion of employees with Bachelor's degree or above | 2.37% | 2.70% | 24.29% | 2.88% | 3.03% | 7.05% |
| Intangible assets | 24.20% | 23.00% | 24.42% | 30.30% | 30.24% | 26.43% |
| Growth rate of intangible assets | 2.01% | 7.00% | 1.48% | 2.54% | 2.48% | 3.10% |
| Income growth rate | 0.74% | 19.36% | 1.40% | 2.08% | 1.88% | 5.09% |
| Net profit growth rate | 3.31% | 0.60% | 0.41% | 1.39% | 0.54% | 1.25% |
| Rate of return on net assets | 0.99% | 0.55% | 0.36% | 0.49% | 0.41% | 0.56% |
| Gross operating margin | 1.72% | 0.86% | 1.12% | 3.56% | 3.01% | 2.05% |

From the weight of each index, it can be seen that the amount of R&D investment and the proportion of R&D personnel are important indicators that affect the level of innovation ability of enterprises, with weights of 19.49% and 10.45%, respectively. The total assets are the amount of R&D investment and the attraction of the talent echelon. Therefore, this index is also one of the important indicators that affect the innovation ability of enterprises, with a weight of 14.53%. In terms of innovation ability output, the net intangible assets and income growth rate are important results of the improvement of enterprise innovation ability, with weights of 26.43% and 5.09%, respectively.

The TOPSIS method is used to calculate the weighted normalization matrix, and the relative closeness of the innovation ability of listed companies in the core area of the Huaihai Economic Zone is obtained. The annual ranking and comprehensive ranking are shown in Table 9. From Table 9, it can be seen that Yankuang ranks first in innovation ability, with a relative closeness of 0.553. The second is Grand Holding, with a relative closeness of 0.3–0.4, followed by Hengrui, Huaibei Mining and XCMG, with a relative closeness of 0.25–0.3. There are 28 companies with relative closeness below 0.1. The last one is LFBC, whose average relative closeness is 0.033, which is far from the first place, indicating that the overall innovation ability of listed companies in the core area of the Huaihai Economic Zone is low and very uneven.

Using SPSS software to carry out k-Means cluster analysis on the innovation ability of listed companies, 37 listed companies are divided into three categories: strong innovation ability, general innovation ability and weak innovation ability. The results are shown in Table 10. According to the clustering results, the innovation ability of listed companies in the core area of the Huaihai Economic Zone is generally weak. There is only one enterprise with strong innovation ability: Yankuang, accounting for 2.7%. The innovation ability of Yankuang Energy ranked second in 2017 and 2018 and then ranked first from 2019 to 2021, indicating that the company's innovation ability is strong and the innovation and development of enterprises is relatively balanced. There are 4 enterprises with general innovation ability: XCMG, Grand Holding, Hengrui, and Huaibei Mining, accounting for 10.8%, and the innovation ability of the remaining enterprises is classified as weak.

Furthermore, although the overall innovation ability of listed companies in this area is weak, there are also many enterprises whose innovation ability is constantly improving from 2017 to 2021, such as Huaibei Mining, Hengrui, Nhwa, Yanghe, Sacred Sun, Shuangxing,

**Table 9. Relative closeness and ranking of the innovation ability of listed companies.**

| Company abbreviation | 2017 | | 2018 | | 2019 | | 2020 | | 2021 | | mean value | rank |
|---|---|---|---|---|---|---|---|---|---|---|---|---|
| | relative closeness | rank | relative closeness | rank | relative closeness | rank | relative closeness | rank | relative closeness | rank | | |
| XCMG | 0.142 | 4 | 0.233 | 5 | 0.246 | 6 | 0.332 | 3 | 0.331 | 4 | 0.257 | 5 |
| Grand Holding | 0.53 | 1 | 0.301 | 3 | 0.268 | 5 | 0.24 | 5 | 0.284 | 5 | 0.325 | 2 |
| Shantui | 0.065 | 11 | 0.067 | 17 | 0.054 | 16 | 0.08 | 18 | 0.075 | 17 | 0.068 | 16 |
| Shenhuo | 0.101 | 7 | 0.114 | 8 | 0.095 | 8 | 0.115 | 8 | 0.117 | 8 | 0.108 | 8 |
| Huafu | 0.063 | 12 | 0.059 | 20 | 0.05 | 18 | 0.054 | 32 | 0.054 | 29 | 0.056 | 24 |
| Sun Paper | 0.066 | 10 | 0.085 | 11 | 0.439 | 2 | 0.097 | 13 | 0.098 | 12 | 0.157 | 6 |
| Ruyi | 0.039 | 23 | 0.04 | 29 | 0.032 | 30 | 0.067 | 23 | 0.049 | 32 | 0.045 | 31 |
| Nhwa | 0.056 | 18 | 0.074 | 15 | 0.042 | 23 | 0.106 | 10 | 0.103 | 9 | 0.076 | 14 |
| Yanghe | 0.079 | 9 | 0.092 | 9 | 0.088 | 9 | 0.117 | 7 | 0.119 | 7 | 0.099 | 10 |
| Yabo | 0.05 | 20 | 0.076 | 14 | 0.049 | 20 | 0.056 | 29 | 0.065 | 22 | 0.059 | 21 |
| LFBC | 0.035 | 30 | 0.035 | 33 | 0.032 | 31 | 0.033 | 37 | 0.029 | 37 | 0.033 | 37 |
| Sacred Sun | 0.037 | 26 | 0.045 | 24 | 0.031 | 33 | 0.051 | 34 | 0.073 | 18 | 0.047 | 30 |
| Shuangxing | 0.034 | 32 | 0.043 | 28 | 0.034 | 28 | 0.067 | 24 | 0.065 | 23 | 0.049 | 28 |
| Limin | 0.061 | 14 | 0.05 | 22 | 0.046 | 21 | 0.078 | 19 | 0.058 | 25 | 0.059 | 23 |
| Fengyuan | 0.037 | 25 | 0.044 | 27 | 0.027 | 37 | 0.042 | 36 | 0.056 | 28 | 0.041 | 34 |
| Liancheng | 0.025 | 34 | 0.035 | 32 | 0.036 | 26 | 0.057 | 28 | 0.043 | 34 | 0.039 | 36 |
| Xiuqiang | 0.035 | 28 | 0.04 | 30 | 0.038 | 25 | 0.055 | 31 | 0.077 | 16 | 0.049 | 27 |
| Yunyi | 0.117 | 6 | 0.129 | 6 | 0.107 | 7 | 0.128 | 6 | 0.12 | 6 | 0.120 | 7 |
| Wuyang | 0.051 | 19 | 0.063 | 18 | 0.039 | 24 | 0.08 | 17 | 0.064 | 24 | 0.059 | 20 |
| Saimo | 0.058 | 16 | 0.088 | 10 | 0.059 | 15 | 0.089 | 14 | 0.073 | 19 | 0.073 | 15 |
| Huaxin | 0.035 | 31 | 0.045 | 25 | 0.029 | 36 | 0.051 | 33 | 0.048 | 33 | 0.042 | 33 |
| Taihe | - | - | - | - | 0.069 | 12 | 0.06 | 27 | 0.05 | 31 | 0.060 | 18 |
| Sidike | - | - | - | - | 0.053 | 17 | 0.085 | 15 | 0.053 | 30 | 0.064 | 17 |
| Yankuang | 0.464 | 2 | 0.508 | 2 | 0.492 | 1 | 0.641 | 1 | 0.659 | 1 | 0.553 | 1 |
| Hengrui | 0.135 | 5 | 0.273 | 4 | 0.312 | 3 | 0.384 | 2 | 0.393 | 2 | 0.299 | 3 |
| VVFB | 0.036 | 27 | 0.05 | 23 | 0.043 | 22 | 0.048 | 35 | 0.038 | 36 | 0.043 | 32 |
| Kanion | 0.062 | 13 | 0.084 | 13 | 0.072 | 11 | 0.111 | 9 | 0.103 | 10 | 0.086 | 12 |
| lukang | 0.046 | 21 | 0.059 | 19 | 0.049 | 19 | 0.073 | 21 | 0.071 | 20 | 0.060 | 19 |
| Hengyuan | 0.101 | 8 | 0.119 | 7 | 0.087 | 10 | 0.104 | 11 | 0.099 | 11 | 0.102 | 9 |
| Huaibei Mining | 0.031 | 33 | 0.511 | 1 | 0.276 | 4 | 0.322 | 4 | 0.336 | 3 | 0.295 | 4 |
| Lianyungang | 0.193 | 3 | 0.07 | 16 | 0.062 | 14 | 0.084 | 16 | 0.082 | 13 | 0.098 | 11 |
| Solareast | 0.06 | 15 | 0.058 | 21 | 0.032 | 32 | 0.065 | 26 | 0.058 | 27 | 0.055 | 25 |
| Cisen | 0.057 | 17 | 0.085 | 12 | 0.063 | 13 | 0.097 | 12 | 0.079 | 14 | 0.076 | 13 |
| KOUZI | 0.044 | 22 | 0.034 | 34 | 0.032 | 29 | 0.075 | 20 | 0.066 | 21 | 0.050 | 26 |
| Quartz | 0.039 | 24 | 0.044 | 26 | 0.035 | 27 | 0.066 | 25 | 0.058 | 26 | 0.048 | 29 |
| Donghong | 0.035 | 29 | 0.037 | 31 | 0.031 | 34 | 0.056 | 30 | 0.043 | 35 | 0.040 | 35 |
| Novoray | - | - | - | - | 0.03 | 35 | 0.068 | 22 | 0.078 | 15 | 0.059 | 22 |

**Table 10. Clustering results.**

| classification | number | Classified enterprises |
|---|---|---|
| Strong innovation ability | 1 | Yankuang |
| general innovation ability | 4 | XCMG,Grand Holding,Hengrui, Huaibei |
| Poor innovation ability | 32 | Shantui,Shenhuo,Huafu,Sun Paper,Ruyi,Nhwa,Yanghe,Yabo,LFBC, Sacred Sun, Shuangxing, Limin,Fengyuan,Liancheng,Xiuqiang,Yunyi, Wuyang,Saimo,Huaxin,Taihe,Sidike,VVFB,Kanion, lukang, Hengyuan,Lianyungang,Solareast,Cisen,KOUZI,Quartz,Donghong,Novoray |

**Table 11. Main obstacle factors of enterprises with strong innovation ability.**

| Yankuang | Obstacle factors | R&D investment | Proportion of employees with Bachelor's degree or above | R&D personnel | Income growth rate | R&D investment intensity |
|---|---|---|---|---|---|---|
| | Obstacle score | 0.3298 | 0.1190 | 0.1065 | 0.0855 | 0.0852 |

Xiuqiang, Yunyi, Kangyuan, Lukang, KOUZI, Novoray and other enterprises. This shows that listed companies pay more attention to the innovation ability of enterprises. However, there are also many enterprises whose innovation ability scores are constantly fluctuating, and innovation investment cannot be continuously invested, which further affects the innovation ability scores of enterprises.

On the basis of the above analysis, the obstacle degree model is used to analyze the main obstacle factors affecting the innovation ability of listed companies. Due to the excessive indicators of the indicator layer, only the top five obstacle factors affecting innovation ability are listed for analysis. The specific results are shown in Tables 11–13.

It can be seen from Table 11 that the top five obstacle factors affecting Yankuang's innovation ability are R&D investment, proportion of undergraduate students, R&D personnel, income growth rate and R&D investment intensity, indicating that innovation investment has a greater impact on the innovation ability of the enterprise. For Yankuang, R&D investment and talent investment have become the first obstacle to the development of the company's capabilities. According to Table 12, the main obstacle factors of XCMG, Grand Holding, Hengrui and Huaibei Mining are slightly different, but the primary obstacle factor is intangible assets, indicating that intangible assets have a great impact on the innovation ability of enterprises. According to Table 13, the top five main obstacle factors of companies with weak innovation ability are intangible assets, R&D investment, total assets, R&D personnel and undergraduate proportion.

In general, the main factors restricting the innovation ability of listed companies in the core area of the Huaihai Economic Zone are intangible assets, R&D investment, total assets, R&D personnel and the proportion of undergraduate students. These factors can be divided into two aspects: capital investment and talent investment. On the one hand, there are fewer listed companies in the region, and the economic development is weak. The treatment that enterprises can provide is difficult to attract more talent to settle down locally, which makes it difficult for enterprises to build a talent echelon. On the other hand, there are few high-tech industries in the region, which are at the bottom of the smile curve of the industrial chain. The

**Table 12. Main obstacle factors of enterprises with general innovation ability.**

| XCMG | Obstacle factors | Intangible assets | R&D investment | Total assets | Proportion of employees with Bachelor's degree or above | R&D personnel |
|---|---|---|---|---|---|---|
| | Obstacle score | 0.2927 | 0.2124 | 0.1221 | 0.0795 | 0.0772 |
| Grand Holding | Obstacle factors | Intangible assets | Total assets | R&D investment | R&D personnel | Proportion of employees with Bachelor's degree or above |
| | Obstacle score | 0.2972 | 0.1595 | 0.1533 | 0.1145 | 0.0772 |
| Hengrui | Oobstacle factors | Intangible assets | R&D investment | Total assets | Proportion of employees with Bachelor's degree or above | Income growth rate |
| | Obstacle score | 0.3073 | 0.2108 | 0.1533 | 0.0810 | 0.0584 |
| Huaibei Mining | Obstacle factors | Intangible assets | R&D investment | Total assets | Proportion of employees with Bachelor's degree or above | R&D investment intensity |
| | Obstacle score | 0.2847 | 0.2396 | 0.1481 | 0.0882 | 0.060 |

**Table 13. Main obstacle factors of enterprises with poor innovation ability.**

| Company abbreviation | Intangible assets | R&D investment | Total assets | R&D investment | Proportion of employees with Bachelor's degree or above |
|---|---|---|---|---|---|
| Shantui | 0.2776 | 0.2057 | 0.1492 | 0.0951 | 0.0736 |
| Shenhuo | 0.2649 | 0.2122 | 0.1290 | 0.1046 | 0.0765 |
| Huafu | 0.2760 | 0.2054 | 0.1442 | 0.0979 | 0.0744 |
| Sun Paper | 0.2844 | 0.2109 | 0.1407 | 0.0980 | 0.0601 |
| Ruyi | 0.2748 | 0.2031 | 0.1492 | 0.1045 | 0.0724 |
| Nhwa | 0.2779 | 0.2043 | 0.1509 | 0.1034 | 0.0723 |
| Yanghe | 0.2779 | 0.2101 | 0.1281 | 0.1063 | 0.0751 |
| Yabo | 0.2752 | 0.2030 | 0.1507 | 0.1083 | 0.0726 |
| LFBC | 0.2721 | 0.2008 | 0.1485 | 0.1059 | 0.0719 |
| Sacred Sun | 0.2747 | 0.2025 | 0.1503 | 0.1048 | 0.0722 |
| Shuangxing | 0.2755 | 0.2032 | 0.1470 | 0.1069 | 0.0731 |
| Limin | 0.2772 | 0.2049 | 0.1511 | 0.1028 | 0.0737 |
| Fengyuan | 0.2740 | 0.2021 | 0.1504 | 0.1081 | 0.0726 |
| Liancheng | 0.2742 | 0.2023 | 0.1504 | 0.1073 | 0.0731 |
| Xiuqiang | 0.2758 | 0.2033 | 0.1510 | 0.1062 | 0.0731 |
| Yunyi | 0.2882 | 0.2123 | 0.1574 | 0.1091 | 0.0766 |
| Wuyang | 0.2758 | 0.2035 | 0.1506 | 0.1055 | 0.0723 |
| Saimo | 0.2777 | 0.2049 | 0.1524 | 0.1058 | 0.0726 |
| Huaxin | 0.2734 | 0.2017 | 0.1503 | 0.1078 | 0.0719 |
| VVFB | 0.2731 | 0.2035 | 0.1481 | 0.1078 | 0.0736 |
| Kanion | 0.2786 | 0.2049 | 0.1511 | 0.1055 | 0.0731 |
| lukang | 0.2761 | 0.2039 | 0.1493 | 0.1013 | 0.0728 |
| Hengyuan, | 0.2790 | 0.2126 | 0.1509 | 0.0875 | 0.0769 |
| Lianyungang | 0.2851 | 0.2094 | 0.1522 | 0.0997 | 0.0755 |
| Solareast | 0.2740 | 0.2033 | 0.1488 | 0.1035 | 0.0723 |
| Cisen | 0.2798 | 0.2059 | 0.1516 | 0.1013 | 0.0735 |
| KOUZI | 0.2741 | 0.2035 | 0.1471 | 0.1084 | 0.0733 |
| Quartz | 0.2757 | 0.2034 | 0.1510 | 0.1072 | 0.0728 |
| Donghong | 0.2745 | 0.2026 | 0.1501 | 0.1071 | 0.0729 |
| Taihe | 0.2766 | 0.2039 | 0.1513 | 0.1083 | 0.0731 |
| Sidike | 0.2771 | 0.2044 | 0.1512 | 0.1080 | 0.0736 |
| Novoray | 0.2754 | 0.2031 | 0.1511 | 0.1087 | 0.0726 |

profit margin of enterprises is low, and there are few funds that can be used to invest in enterprise innovation, which further restricts the improvement of enterprise innovation ability.

In summary, this paper uses the entropy method and entropy-topsis model to evaluate the innovation ability of listed companies in the core area of the Huaihai Economic Zone and comprehensively analyzes the influencing factors of the innovation ability of listed companies. From the results, the ranking order of the evaluation results of the two methods is slightly different but basically the same. For example, Yankuang Energy, Hengrui and Huaibei Mining are excellent enterprises with high investment and high output, and the evaluation of innovation ability is in the forefront; although Xugong Machinery and Yuanda Holdings belong to the high input and low output quadrant of innovation, due to the relatively high input and output of innovation, the evaluation of innovation ability is also ranked at the top. Of course, the entropy-topsis model more specifically gives the innovation ability score of each enterprise and the specific obstacle factors affecting the innovation ability and supplements the evaluation results of the entropy method.

Through the combination of the above two methods, this paper comprehensively and accurately evaluates the innovation ability of listed companies in the core area of the Huaihai Economic Zone. Of course, due to the limited information disclosed by listed companies, there are few selection indicators, the R&D cycle of different industries is quite different, and the innovation output and innovation input are not synchronized. Therefore, the data analysis in this paper may deviate slightly from the comprehensive and objective evaluation of the innovation ability of listed companies.

## Conclusions and suggestions

### Conclusions

Based on the research and analysis of the innovation capabilities of listed companies in the core economic region of Huaihai from 2017 to 2021, this paper draws the following conclusions:

First, the innovation ability of listed companies in the Huaihai core economic zone is weak and different. First, there are only 37 listed companies in the eight cities in the region, the number of listed companies is small, and the number of listed companies in each city is unbalanced. The number of listed companies in Xuzhou and Jining is the largest, with 9 each, and Shangqiu has only one. Second, the overall innovation ability of listed companies in the region is weak. There is only 1 listed company with strong innovation ability in the whole region, 4 companies with general innovation ability, 86.5% of enterprises with weak innovation ability, and only 12% of enterprises with high investment and high output. Third, the innovation ability of listed companies in this region varies greatly, and their development is uneven. According to the ranking of the innovation ability score, the highest score is 0.553, and the lowest score is 0.033. From the perspective of subinnovation ability, the average innovation input has the highest score of 4.65 and the lowest score of 2.28, and the innovation output has the highest score of 25.6 and the lowest score of 20.49. There are obvious differences in the innovation ability of enterprises in this region.

Second, the main reason for the lack of innovation ability of listed companies in the Huaihai economic core area is the lack of capital investment and talent investment. According to the data, 65% of enterprises are in the low input quadrant. The fundamental reason for the low innovation ability of enterprises in the region is the lack of innovation investment. The main obstacle factors are intangible assets, R&D investment, total assets, R&D personnel and undergraduate ratio. Although enterprises in the region have increased their investment in scientific and technological innovation in recent years, the proportion of R&D personnel has increased year by year with the introduction of highly educated talent, and the overall investment in enterprise innovation has shown an upward trend. However, it should be noted that up to 86.5% of enterprises belong to the category of weak innovation ability, so attention should be given to continuing to increase investment in innovation to provide guarantees for the improvement of innovation ability. In addition, in recent years, the innovation output of each company has also improved, but the growth rate of income and net profit has not increased significantly, and the innovation output is still not high. The main reason is the lack of capital investment and talent investment in R&D personnel and R&D investment, which directly restricts the output of enterprise innovation results, resulting in a low level of overall innovation ability of enterprises. Therefore, the innovation input of listed companies in the Huaihai Economic Core Area needs to be strengthened, and the innovation output needs to be improved.

Third, the innovation primacy of Xuzhou listed enterprises is not high. As the central city of the Huaihai economic zone. As the central city of the Huaihai economic zone, the

innovation primacy of Xuzhou listed enterprises is not high. First, from the analysis of the number of listed companies, the number of listed companies in Xuzhou and Jining is equal, ranking first. Therefore, it is necessary to support the cultivation of more listed companies and enhance the status of regional centers. Second, from the analysis of innovation input, although the overall innovation input of Xuzhou ranks first in the area, compared with other cities, the advantages are not obvious, not much different, and need further improvement. Third, from the analysis of innovation capacity output and innovation input–output ratio, the ranking is not in the forefront and does not have an advantage. Finally, from the perspective of overall innovation ability, the innovation ability of Xuzhou listed companies is not high and unbalanced. The top XCMG and Yunyi in Xuzhou's innovation capability ranking are ranked 5 and 7, respectively. Of course, due to the long input–output cycle of XCMG, Yunyi and other enterprises, the advantages are not obvious. Nhwa, Seymour, Wuyang and Limin ranked 14, 15, 20 and 23, respectively. There are also VVFB, Huaxin, and LFBC innovation abilities ranked after 30. Therefore, Xuzhou city should constantly improve its innovation leading force.

## Suggestions

According to the research conclusions, the relevant recommendations for listed companies, governments and central cities are as follows.

First, companies should pay attention to the improvement of innovation ability. Only through continuous innovation can enterprises survive better in the future. The key factor in improving innovation ability is to attach importance to innovation input, which could constrain innovation output and ultimately affect the improvement of innovation ability. Therefore, enterprises should not only increase investment in R&D funds but also pay attention to the quantity and quality of R&D personnel. Companies can make relevant talent introduction plans and take advantage of profitable conditions to attract highly educated talent to enrich and expand the R&D team. Government departments should also increase the effort of talent introduction at the government level to draw policy and further attract talent so that under the joint efforts of the government and enterprises, the innovation ability of the region will be improved, and the economic development potential of the Huaihai economic zone can be revitalized.

Second, the local government should formulate relevant supporting policies to encourage enterprise innovation and create a good environment for innovation. The listed companies in the Huaihai economic area are mainly involved in engineering machinery, food, medicine, mining and energy, biochemicals, manufacturing and other industries. The overall innovation ability of enterprises is not strong, and there are few science and technology enterprises. The innovation ability of emerging industries needs to be strengthened. Therefore, the government needs to create a good innovation environment to support enterprise innovation. The government should support listed companies with strong innovation ability to maintain innovation, research and development efforts, support listed companies in the traditional manufacturing industry to accelerate the pace of transformation and upgrading, and focus on cultivating high-tech companies with innovative growth potential to actively go public.

Third, Xuzhou city should constantly improve the innovation leading force. First, Xuzhou should give full play to its industrial advantages and industrial characteristics and insist on giving full play to the advantages and characteristics of the first industry represented by XCMG machinery. Second, Xuzhou city should fully integrate resources to optimize its industrial structure. By providing policy, financial support and guidance, Xuzhou city should cultivate leading high-quality companies and increase the number of listed companies. At the same

time, by building an advanced manufacturing base with international competitiveness, an industrial science and technology innovation center and a national modern agriculture demonstration zone, we will vigorously develop and cultivate high-tech enterprises to promote the optimization and upgrading of Xuzhou's industrial structure and social and economic development.

## Supporting information

**S1 Data.**
(DOCX)

## Author Contributions

**Formal analysis:** Qin-Xia Liu.

**Investigation:** Qin-Xia Liu.

**Methodology:** Qin-Xia Liu.

**Project administration:** Qin-Xia Liu.

**Software:** Qin-Xia Liu.

**Writing – original draft:** Qin-Xia Liu.

**Writing – review & editing:** Qin-Xia Liu.

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
