## [Decision Letter · Decision Letter 0]

16 Jan 2023

PONE-D-22-34910Research on the Innovation Ability of Different Listed Companies in the Core Area of the Huaihai Economic ZonePLOS ONE

Dear Dr. Liu,

Thank you for submitting your manuscript to PLOS ONE. After careful consideration, we feel that it has merit but does not fully meet PLOS ONE’s publication criteria as it currently stands. Therefore, we invite you to submit a revised version of the manuscript that addresses the points raised during the review process.

The reviewers have suggested minor revision, please see attached / below this email. Once you login to your account, you'll be able to see the comments and suggestions. 

We look forward to receiving your revised manuscript.

Kind regards,

Muhammad Khalid Bashir, PhD

Academic Editor

PLOS ONE

Journal Requirements:

" This work was supported by the Xuzhou Social Science Research Project, No. 22XSM-167, Study on benchmarking gap and promotion path between Xuzhou and national central cities; Project of Jiangsu Vocational Institute of Architectural Technology, No. JYA320-24, Evaluation and comparative study on the primacy of scientific and technological innovation in Xuzhou from the perspective of value chain."

"This work was supported by the Xuzhou Social Science Research Project, No. 22XSM-167, Study on benchmarking gap and promotion path between Xuzhou and national central cities; Project of Jiangsu Vocational Institute of Architectural Technology, No. JYA320-24, Evaluation and comparative study on the primacy of scientific and technological innovation in Xuzhou from the perspective of value chain."

" This work was supported by the Xuzhou Social Science Research Project, No. 22XSM-167, Study on benchmarking gap and promotion path between Xuzhou and national central cities; Project of Jiangsu Vocational Institute of Architectural Technology, No. JYA320-24, Evaluation and comparative study on the primacy of scientific and technological innovation in Xuzhou from the perspective of value chain."

Additional Editor Comments:

Please incorporate the suggested changes.

Reviewers' comments:

Reviewer's Responses to Questions

**Comments to the Author**

1. Is the manuscript technically sound, and do the data support the conclusions?

Reviewer #1: Yes

Reviewer #2: Yes

2. Has the statistical analysis been performed appropriately and rigorously? 

Reviewer #1: Yes

Reviewer #2: Yes

3. Have the authors made all data underlying the findings in their manuscript fully available?

Reviewer #1: Yes

Reviewer #2: Yes

4. Is the manuscript presented in an intelligible fashion and written in standard English?

Reviewer #1: Yes

Reviewer #2: Yes

5. Review Comments to the Author

Reviewer #1: Title

1. Instead of using word “Research” use some other appropriate word describing the main objective of the study such as “Analyzing the innovative ...” Or

2. “Identifying determinants of innovative……..” or “Factors affecting innovative …….” as at end of the introduction while describing objectives author(s)

Abstract

1. Add the importance/justification of the study i.e. why this research has been undertaken in the beginning and then describe the objectives of the study.

2. Add importance/justification the choice of the “Entropy Weight Analysis” method and description of source of the data collection.

3. In the last 2-3 sentences of the abstract, actual results, concluding statement and main recommendation(s) should be provided.

Introduction

1. Justification of the study is lacking. Why the objectives of the study are important for conducting this research. Problem statement should clearly be defined and objectives should be described more clearly.

2. Avoid long sentences and use proper punctuation, commas and semicolons, to separate the parts of a sentence so that reader can easily understand what the author intends to say.

Review of Literature

1. All/most of the review should be comprehended. Review of about nine studies is added while Author(s) stated in the beginning of the last paragraph of the review of literature that comprehensive research is available and also mentioned the various methods e.g. factor analysis. So, all such literature should be comprehended.

2. If possible, provide the conceptual framework of the study

Methodology (Study Design)

1. Instead of study design as a main heading, heading such as methodology or data and methods etc can be given and study design could be the sub-headings

2. Write some intro of study design before the heading “Construction of the innovation capability evaluation index system” to depict what is going to be written in upcoming sub-headings

3. A heading, “Theoretical Background”, may be added for presenting theories used / theoretical literature.

4. Data collection and analytical methods (Entropy Method) should be backed by the literature and justification should be provided that why this is preferred over other methods such as factor analysis widely used in the literature.

Results and Discussion (Evaluation of the innovation ability of listed companies in the core area of the Huaihai economic zone)

1. Instead of “Evaluation of the innovation ability of listed companies in the core area of the Huaihai economic zone”, Results and Discussion” heading may be used and sub-headings under this main headings could be used

2. Table 6 onward and figure 2 onward, only the findings re described and expected reasons behind these findings are lacking. It is suggested that some logics behind these findings should be added

3. Top companies with high input and high output should be explained well so that they can serve as role model.

4. Comparison of this study with the earlier studies is not added. Such discussion must be added so that reader can have a clear idea that how the findings of this study support or contradict the findings of earlier studies.

Suggestions

Suggestions must come from the findings of the data analysis of the study. In this regard, improve suggestion 2 and 3.

Advantages and Disadvantages

1. Advantages and Disadvantages of what?

2. Is it a proper place for this heading at end of suggestion

3. Find more appropriate place in introduction or in methodology or results and discussion

Reviewer #2: This paper is a good effort to analyze innovation ability of companies in the Huaihai region. Methodolgoy of the entropy method is appropriate to address the objective of this study. In upcoming research on this aspect relying on just one method (entropy method) would not suffice. A mix method approach would provide better insight.

6. PLOS authors have the option to publish the peer review history of their article (what does this mean?). If published, this will include your full peer review and any attached files.

Reviewer #1: No

Reviewer #2: No

---

## [Author Response · Author response to Decision Letter 0]

3 Mar 2023

Reviewer #

Title

1. Instead of using word “Research” use some other appropriate word describing the main objective of the study such as “Analyzing the innovative ...” Or

2. “Identifying determinants of innovative……..” or “Factors affecting innovative …….” as at end of the introduction while describing objectives author(s)

The title has been revised in accordance with these recommendations as follows: “Analyzing the innovation ability of listed companies in the core area of the Huaihai economic zone.”

Abstract

1. Add the importance/justification of the study i.e. why this research has been undertaken in the beginning and then describe the objectives of the study.

2. Add importance/justification the choice of the “Entropy Weight Analysis” method and description of source of the data collection.

3. In the last 2-3 sentences of the abstract, actual results, concluding statement and main recommendation(s) should be provided.

According to these recommendations, the abstract has been modified in three ways. First, the significance and the objective of the research have been highlighted. Second, the selection of methods were included, and the sources of data collection have been added. Third, conclusions and main recommendations have been provided at the end of the abstract.

Introduction

1. Justification of the study is lacking. Why the objectives of the study are important for conducting this research. Problem statement should clearly be defined and objectives should be described more clearly.

2. Avoid long sentences and use proper punctuation, commas and semicolons, to separate the parts of a sentence so that reader can easily understand what the author intends to say.

 In accordance with these recommendations, the introduction has been revised. A statement of the significance and objectives of the study has been added, and the language and content have been polished.

Review of Literature

1. All/most of the review should be comprehended. Review of about nine studies is added while Author(s) stated in the beginning of the last paragraph of the review of literature that comprehensive research is available and also mentioned the various methods e.g. factor analysis. So, all such literature should be comprehended.

2. If possible, provide the conceptual framework of the study

 The literature review has been polished and modified to make the content easier to understand; it also provides a conceptual framework for the research.

Methodology (Study Design)

1. Instead of study design as a main heading, heading such as methodology or data and methods etc can be given and study design could be the sub-headings

 The main heading has been revised as suggested.

2. Write some intro of study design before the heading “Construction of the innovation capability evaluation index system” to depict what is going to be written in upcoming sub-headings

 According to the recommendations, the content about the research design has been revised, and the sample sources and research methods have been introduced.

3. A heading, “Theoretical Background”, may be added for presenting theories used / theoretical literature.

 The Theoretical Background heading has been added as suggested.

4. Data collection and analytical methods (Entropy Method) should be backed by the literature and justification should be provided that why this is preferred over other methods such as factor 

Because the research object of this paper is panel data, the entropy method is more suitable than the factor analysis method. In addition, to conduct more comprehensive and scientific research, this paper also adds the TOPSIS-Entropy combined method to compare and verify the previous analysis results.

Results and Discussion (Evaluation of the innovation ability of listed companies in the core area of the Huaihai economic zone)

1. Instead of “Evaluation of the innovation ability of listed companies in the core area of the Huaihai economic zone”, Results and Discussion” heading may be used and sub-headings under this main headings could be used

 The heading has been revised as suggested.

2. Table 6 onward and figure 2 onward, only the findings re described and expected reasons behind these findings are lacking. It is suggested that some logics behind these findings should be added

 The survey results and the expected reasons for these findings have been added in accordance with these recommendations.

3. Top companies with high input and high output should be explained well so that they can serve as role model.

 According to these recommendations, the analysis of typical model enterprises with high input and high output has been further explained (e.g., Hengrui).

4. Comparison of this study with the earlier studies is not added. Such discussion must be added so that reader can have a clear idea that how the findings of this study support or contradict the findings of earlier studies.

 Due to the lack of research on the innovation ability of the listed enterprises in the Huaihai Economic Zone, there is no way to compare our research with earlier studies. Therefore, this paper adds the TOPSIS-Entropy combined method as a comparative supplementary analysis method to evaluate the Huaihai Economic Zone more scientifically and comprehensively.

Suggestions

Suggestions must come from the findings of the data analysis of the study. In this regard, improve suggestion 2 and 3.

This part has been modified and improved according to these suggestions.

Reviewer #2: This paper is a good effort to analyze innovation ability of companies in the Huaihai region. Methodolgoy of the entropy method is appropriate to address the objective of this study. In upcoming research on this aspect relying on just one method (entropy method) would not suffice. A mix method approach would provide better insight.

The paper has been modified and improved according to these suggestions, and the TOPSIS-Entropy combined method has been added.

---

## [Decision Letter · Decision Letter 1]

18 Apr 2023

Analyzing the innovation ability of listed companies in the core area of the Huaihai economic zone

PONE-D-22-34910R1

Dear Dr. Liu,

We’re pleased to inform you that your manuscript has been judged scientifically suitable for publication and will be formally accepted for publication once it meets all outstanding technical requirements.

Kind regards,

Muhammad Khalid Bashir, PhD

Academic Editor

PLOS ONE

Additional Editor Comments (optional):

Reviewers' comments:

Reviewer's Responses to Questions

**Comments to the Author**

1. If the authors have adequately addressed your comments raised in a previous round of review and you feel that this manuscript is now acceptable for publication, you may indicate that here to bypass the “Comments to the Author” section, enter your conflict of interest statement in the “Confidential to Editor” section, and submit your "Accept" recommendation.

Reviewer #1: (No Response)

Reviewer #2: All comments have been addressed

2. Is the manuscript technically sound, and do the data support the conclusions?

Reviewer #1: Yes

Reviewer #2: Yes

3. Has the statistical analysis been performed appropriately and rigorously? 

Reviewer #1: Yes

Reviewer #2: Yes

4. Have the authors made all data underlying the findings in their manuscript fully available?

Reviewer #1: (No Response)

Reviewer #2: Yes

5. Is the manuscript presented in an intelligible fashion and written in standard English?

Reviewer #1: Yes

Reviewer #2: Yes

6. Review Comments to the Author

Reviewer #1: (No Response)

Reviewer #2: (No Response)

7. PLOS authors have the option to publish the peer review history of their article (what does this mean?). If published, this will include your full peer review and any attached files.

Reviewer #1: No

Reviewer #2: **Yes: **Tahira Sadaf

---

## [Editor Report · Acceptance letter]

20 Apr 2023

PONE-D-22-34910R1 

Analyzing the innovation ability of listed companies in the core area of the Huaihai economic zone 

Dear Dr. Liu:

I'm pleased to inform you that your manuscript has been deemed suitable for publication in PLOS ONE. Congratulations! Your manuscript is now with our production department. 

Kind regards, 

on behalf of

Dr. Muhammad Khalid Bashir 

Academic Editor

PLOS ONE